# Clinical and Biomarker Profile Responses to Rehabilitation Treatment in Patients with Long COVID Characterized by Chronic Fatigue

**DOI:** 10.3390/v15071452

**Published:** 2023-06-27

**Authors:** Jessica Binetti, Monica Real, Marcela Renzulli, Laia Bertran, David Riesco, Carles Perpiñan, Alba Mohedano, Rosa San Segundo, Marta Ortiz, José Antonio Porras, Daniela Rosanna Pineda, Teresa Auguet

**Affiliations:** 1GEMMAIR Research Group, Department of Medicine and Surgery, Pere Virgili Institute for Health Research (IISPV), Rovira i Virgili University (URV), 43007 Tarragona, Spain; jessica.binetti@gmail.com (J.B.); laia.bertran@urv.cat (L.B.); driesco.hj23.ics@gencat.cat (D.R.); aporras.hj23.ics@gencat.cat (J.A.P.); 2Internal Medicine Unit, Joan XXIII University Hospital of Tarragona, 43007 Tarragona, Spain; mreal.hj23.ics@gencat.cat (M.R.); amohedano.hj23.ics@gencat.cat (A.M.); 3Rehabilitation Unit, Joan XXIII University Hospital of Tarragona, 43007 Tarragona, Spain; mrenzulli.hj23.ics@gencat.cat (M.R.); rsansegundo.hj23.ics@gencat.cat (R.S.S.); martaoa.green@gmail.com (M.O.); danipineda89@hotmail.com (D.R.P.); 4Catalan Institute for Health (ICS), 43202 Reus, Spain; cperpinan.tgn.ics@gencat.cat; 5Neurobehavioral and Health Research Group (NEUROLAB), Rovira i Virgili University (URV), 43007 Tarragona, Spain

**Keywords:** long COVID, chronic fatigue, physical rehabilitation, biomarkers, SARS-CoV-2

## Abstract

Long COVID (LC) syndrome is a complex multiorgan symptom that persists beyond >12 weeks after SARS-CoV-2 infection. The most frequently associated symptom is fatigue. Physical activity and exercise are recommended, although specific studies are lacking. The objectives of the present work are to analyze the impact of a supervised exercise program on the clinical evolution of LC with fatigue patients and to identify whether certain circulating biomarkers could predict the response to rehabilitation. The rehabilitation treatment response was analyzed in 14 women diagnosed with LC and fatigue, based on the changes in the 6 min walk test and Borg/Fatigue Impact scales. Patients who showed improvement in the meters walked were considered “responders” to the therapy. A total of 65% of patients responded to the exercise program, with an improvement in the meters walked and in oxygen saturation, with stability in the percentage of meters walked. Participants with obesity and those double-vaccinated against SARS-CoV-2 presented a lower degree of fatigue. LC patients presented a favorable response to a supervised exercise program. Differences in creatinine and protein levels were observed between rehabilitation therapy “responders” and “nonresponders”. A good state of protein nutrition was related to a better rehabilitation response. The results are promising regarding possible predictive biomarkers of rehabilitation response, such as creatinine.

## 1. Introduction

Coronavirus disease 19 (COVID-19) is an infectious disease caused by the severe acute respiratory syndrome coronavirus type 2 (SARS-CoV-2). Most people with COVID-19 experience mild-to-moderate illness, while approximately 10–15% develop severe illness and 5% become critically ill [1]. The average recovery time from COVID-19 is 2–3 weeks depending on symptomatology [2,3,4]. However, between 20–90% of patients may exhibit symptoms for several weeks or months [5,6,7,8,9]. This situation is named post-acute sequelae of COVID-19, also now known as long COVID (LC), which is defined as the persistence of symptoms for >12 weeks or new symptoms attributable to COVID-19 [10]. Common complaints are fatigue, dyspnea, brain fog, orthostatic intolerance, and some systemic illnesses [11]. At least 65 million individuals worldwide are estimated to have LC, with cases increasing daily [12]. Specifically, the prevalence of LC reported in our geographical area, Spain, has been up to 48% [9].

LC is a complex multisystemic/multifactorial disorder with a not yet fully understood pathogenesis. However, based on the history of viral inflammatory diseases and other research evidence on SARS-CoV-2, it is suggested that the seeding and persistence of SARS-CoV-2 in different organs and its reactivation, and the immune response to unrelated viruses, autoimmunity, microbiota disruption, and uncontrolled inflammation, along with thromboembolism, lung dysfunction, and nervous system dysfunction due to occult neuronal injury during SARS-CoV-2 infection are major drivers of LC. This complex pathophysiology likely drives different clinical phenotypes [11,13,14,15].

Regarding weakness/fatigue and neurologic symptomatology, it appears that the combination of an exaggerated production of inflammatory cytokines, the demyelinating response in the central nervous system, and the role of the altered adaptive immune response could explain the symptoms of chronic muscle weakness, sensory abnormalities, or cognitive and autonomic dysfunction seen in LC. Another hypothesis underlying fatigue LC pathogenesis is a biochemical alteration of critical mitochondrial metabolic pathways that could produce apoptosis of muscle cells [13].

Fatigue is one of the most frequent extra-respiratory symptoms of SARS-CoV-2 infection, described in 41.4% of the patients included in the largest published cohort [16]. Regarding persistent fatigue, data published in two studies suggest a frequency of 35–53% at 4–8 weeks post-infection and 16% at 12 weeks post-infection [17,18]. In other studies, this frequency is higher, up to 51% [7,8,19,20]. The nature of fatigue in COVID-19 patients shares features with chronic fatigue after other infections, such as SARS, MERS, and community-acquired pneumonia.

Some authors have studied predictors of LC. Older age groups, underlying comorbidities and some abnormal laboratory results seemed to make COVID-19 patients more susceptible to developing LC [20]. More recently, Sudre et al. described that early disease features could be predictive of its duration. With only three features (symptoms in the first week, age, and sex), they built a model designed to stratify patients into short (<10 d) and long (≥28 d) durations of COVID-19. The authors suggested that this model could be used to identify individuals at risk of LC for trials of prevention or treatment and to plan education and rehabilitation services [3]. However, although predictive biomarkers of the appearance of LC after a SARS-CoV-2 infection have been described [14,21], there are no studies of biomarkers of response to rehabilitation treatment in patients affected by LC fatigue. Specifically, fatigue in patients with LC has a great impact on their mental health and on their ability to resume their social life and return to work [22]. Therefore, the economic and social consequences are considerable. The only treatment currently considered useful is a specific rehabilitation program [23,24]. However, not all patients respond adequately to rehabilitation interventions, so knowing in advance those most likely to respond would be very interesting in terms of designing efficient and personalized treatment programs. In this sense, the present study aimed to describe the response to rehabilitation treatment in patients with LC with persistent fatigue as the main symptom and also to analyze whether certain blood biomarkers could predict the response to rehabilitation treatment in a cohort from a geographical area with a high prevalence of this pathology.

## 2. Materials and Methods

### 2.1. Subjects

This study was approved by the institutional review board, and all participants gave written informed consent before taking part in the study (156/2021). A longitudinal prospective intervention pilot study was carried out, with a before–after analysis of the clinical response to the standardized and individualized rehabilitation program.

The study period was from September 2021 to May 2022. The study population consisted of 14 women, identified and recruited in the long COVID monographic consultation (LCMC) at the Hospital Universitari de Tarragona Joan XXIII (HJ23), diagnosed with LC and with predominant symptoms of fatigue.

### 2.2. Data Collection

As shown in Figure 1, the patients were first visited in the consultation of LCMC, and later they were referred to the consultation of Rehabilitation. In both consultations, the participants underwent complete anamnesis and physical, anthropometric, and biochemical evaluations to obtain clinical and analytical data. Additionally, the patients were given the Fatigue Impact Scale (FIS) and the Borg scale to assess fatigue and dyspnea, respectively. A first 6 min walk test (6MWT) was performed on all of them. If the result of 6MWT was >85% of the predicted value, the patient was referred home with WHO’s global recommendations on physical activity form health [25]. Bearing in mind that a 6MWT result >85% indicates good walking ability that does not require rehabilitation intervention. If 6MWT <85%, they were referred to carry out the specific rehabilitation program in the rehabilitation service. Before starting the rehabilitation sessions, desaturation due to exertion and heart failure after COVID-19 were ruled out. Subjects were included in a supervised exercise program of 12 to 20 physiotherapy sessions for three months. In each session, an initial warm-up phase was carried out with stretching and light exercise (5 to 10 min), followed by aerobic exercise on a bicycle, of progressive duration (10 to 30 min) and intensity according to effort tolerance, with heart rate monitoring. In one of the weekly sessions, muscle strengthening exercises for the upper extremities were performed. In addition, the patient was encouraged to perform an aerobic routine (walking) on the days that he/she did not come to the hospital. At the end of the guided hospital exercise program, a guideline was given to continue their practice at home. At the end of the treatment, a second 6MWT was performed, and the patients were classified into Group 1 “responders” and Group 2 “non-responders”. The patients who showed improvement in the meters walked were considered responders to the rehabilitation therapy. Moreover, all the patients were contacted by telephone to obtain data on the perception of fatigue at the end of the rehabilitation treatment (FIS) and their global satisfaction with participation in the study.

In the face-to-face visits at the LCMC, the health personnel collected: clinical data from the participants, symptoms of acute COVID-19 infection, radiological findings, hospital admission and analytical data and treatment received, current symptoms of LC, physical examination constants (blood pressure, O2 saturation, heart rate (HR), and respiratory rate), cardiopulmonary, abdominal, and neurological examination, and body mass index (BMI), electrocardiogram (ECG), neuropsychological screening using the Hospital Anxiety and Depression (HAD) scale and Montreal cognitive assessment (Moca scale) [26]. The evaluation in the RHBC, the initial physical condition was evaluated, measuring fatigue and other symptoms through the 6 MWT; then the initial and final constants (systolic blood pressure (SBP), cardiac frequency, peripheral oxygen saturation) were collected and the initial and final Borg dyspnea and fatigue scale scores (FIS) were evaluated, as well as the number of meters walked and the percentage of predicted value (calculated using the reference equation for 6MWT appropriate to the age, height, weight, and sex of the patient).

### 2.3. Biochemical Analyses

All of the subjects included underwent physical, anthropometric, and biochemical assessments. Blood samples were obtained from the patients at the time of the initial assessment in the LCMC for basic hematological and biochemical analysis, serological profile, and determination of the levels of proinflammatory molecules (interleukin 6 (IL-6) and C-reactive protein (CRP)) and autoimmune molecules (antinuclear antibodies (ANA), lupus anticoagulant (AcL) and beta-2-glycoprotein I (B2GPI)), using conventional automated analyzers after 12 h of fasting.

### 2.4. Rehabilitation Intervention and Evaluation

Due to the nature and duration of the intervention, it was not possible to blind the research team or the participants to the treatment allocation. After performing the 6MWT at the initial visit to the RHBC, two groups of patients were identified according to the percentage of meters walked, according to Casanova et al. [27]. As described, those with 6MWT > 85% carried out a set of rehabilitation exercises at home (aerobic exercise), while subjects with 6MWT < 85% were included in a supervised exercise program in the hospital rehabilitation service, similar to the one described by Cattadori et al. [28]. It is important to note that the patients did not receive any new medication or related supplements coinciding with the rehabilitation program.

Once the rehabilitation therapy was carried out, the response to it was evaluated by comparing the initial and final data of the 6 MWT. The difference in the walked meters was used to classify the patients as favorable evolution (“responder” patients) or unfavorable (“nonresponder”). We also valued differences in relation to the percentage of meters, the fatigue score (FIS), and the oxygen saturation percentage. Finally, we also analyzed the response to the rehabilitation treatment according to the presence of obesity (obese: body mass index (BMI) > 30 Kg/m^2^; non-obese: BMI < 30 Kg/m^2^).

### 2.5. Statistical Analysis

The data were analyzed using the SPSS/PC+ for Windows statistical package (version 23.0; SPSS, Chicago, IL, USA). The Kolmogorov-Smirnov test was used to assess the distribution of variables. Continuous variables are reported as the mean (SD); noncontinuous variables are reported as the median and 25th–75th percentile. Categorical variables are expressed as absolute frequencies or percentages. The comparative analyses were performed using Student’s *t*-test for parametric variables or a nonparametric Mann-Whitney U test. Regarding the categorical variables, significant differences between two groups were assessed using the chi-square test. The strength of the association between variables was calculated using Spearman’s rho correlation test. *p* values < 0.05 were considered statistically significant.

## 3. Results

### 3.1. Baseline Characteristics of Subjects

The baseline characteristics of patients and features of acute COVID-19 are given in Table 1. 

It is worth mentioning that of the 14 patients initially included, 5 had a result of 6MWT >85% and were referred home with recommendations for basic exercise. One of the participants was lost to follow-up before carrying out the RHBC, so she could not be classified by treatment. The rest of them entered the rehabilitation program of the hospital, and after rehabilitation treatment and with the results of the second 6MWT, were classified into Groups 1 (“responders”) and 2 (“non-responders”).

The cohort had a mean age of 44.21 years and comprised all women. Only 35.7% presented comorbidities, and 21.4% had allergies. No patient presented normal weight, half of the sample was overweight (50%), and 92.1% of the sample was between overweight and grade II obesity. A total of 78.6% of the participants exhibited a low level of baseline physical activity.

Regarding the COVID-19 disease picture, all of the patients were diagnosed with COVID-19 between 27 October 2020, and 1 August 2021, in the context of the Delta variant. In the total cohort, no patient was asymptomatic, 64.3% presented mild/moderate clinical symptoms, and 35.7% showed severe symptoms requiring hospital admission. A total of 35.7% of the subjects presented pathological findings on chest X-rays. Regarding vaccination, 78.6% of patients were vaccinated, 28.6% with a single dose, and 50% with a booster dose. As a treatment, 100% of participants required symptomatic treatment, 42.9% required antibiotic treatment, 35.7% required treatment with corticosteroids, 35.7% required treatment with low molecular weight heparin, and 7.1% (n = 1) required treatment with remdesivir. There were no candidates for tocilizumab.

The subjects who had required hospital admission for acute COVID-19 (n = 5) presented the analytical results of admission shown in Table 2, highlighting a mean of the sample above the reference values for aspartate aminotransferase (AST, 71.40 (68.30)) and alanine aminotransferase (ALT, 89.80 (78.20)), C-reactive protein (CRP, 2.48 (2.00)), D-dimer (759.50 (458.90)) and interleukin (IL)-6 (24.26 (14.60)). These values were clearly different from those of the laboratory tests carried out in the LCMC, although these differences were not statistically significant, probably due to the small number of patients who had required hospital admission. Nevertheless, protein levels significantly increased in the LCMC. During the study, two patients became reinfected.

#### Long COVID Syndrome

The four symptoms with the highest frequency of appearance in the participants (n = 14) were fatigue (since its presence was a study inclusion criterion), headache, dyspnea, and amnesia (Figure 2). The participants presented a mean of 4.07 (1.70) symptoms, and 35.7% of participants manifested more than 4 symptoms.

### 3.2. Clinical Assessment

#### 3.2.1. Long COVID Monographic Consultation

The data collected in the initial assessment carried out using the LCMC are summarized in Table 3. 

The data collected during the consultation remained within normal ranges and the physical examination was normal except for 21.4% of the subjects (the cardiorespiratory auscultation of two patients and the abdominal examination of one patient were pathological). All of the electrocardiograms were normal.

In reference to the analysis requested in the consultation, the values of fibrinogen (445.58 (57.90)) and glomerular sedimentation rate (GSR) (14.92 (8.40)) were increased. A total of 16.7% of the patients presented some markers of autoimmunity, two patients were positive for ANA (titers 1/320 and 1/640), and one was positive for AcL. Antibodies against SARS-CoV-2 were detected in all subjects. When comparing the values of the laboratory data on admission with the values of the analytics performed in the consultation, we found no significant differences. As shown in Table 2, it can be observed that the values of AST, ALT, CRP, and D-dimer at admission were much higher than those of the laboratory tests performed months later, although without significant differences. In the neuropsychological screening, it was observed that 57.1% of the participants had anxiety and 14.3% had depression.

#### 3.2.2. Rehabilitation Consultation

As mentioned, five patients did not enter the hospital rehabilitation program and were sent home with recommendations for basic physical exercise. Nine patients started a hospital rehabilitation program. When we compared both groups in relation to pathological history, clinical characteristics, and current analytical parameters, we did not find any significant differences.

The data obtained in the 6MWT performed in the RHBC before and after carrying out the rehabilitation treatment are detailed in Table 4. It is important to note that the parameters measured in the first visit to the RHBC (before receiving rehabilitation treatment) and in the second visit (after receiving it) were within normal limits. Regarding the meters walked in the initial test, the participants covered an average of 431.92 (58.90) meters in 6 min, and the average percentage of meters covered was 78.50 (9.90).

### 3.3. Response to Treatment

When we evaluated the response to treatment, we found no significant differences in the Borg scale for dyspnea or FIS for fatigue before and after receiving rehabilitation treatment. However, regarding the 6 MWT, we found a significant improvement in the meters walked before and after the rehabilitation intervention. Moreover, we found stability in the percentage of meters walked (Table 4).

SBP was significantly lower than in the initial test. The rest of the variables did not show significant differences.

Finally, at the end of the rehabilitation treatment, through a telephone call, fatigue and satisfaction scales were collected. The fatigue impact scale presented a mean score of 22.14 (4.90) (out of 32), and the participants gave a mean score of 8.36 (1.50) out of 10 for their satisfaction with the study.

### 3.4. Differential Characteristics of “Responders” and “Nonresponders” Patients

After the rehabilitation treatment, we classified the patients according to the improvement in walked meters as “responders” and “nonresponders”, and we compared the main characteristics between both groups. We did not observe any difference in the clinical data collected (age, allergies, comorbidities, level of physical activity, characteristics of acute COVID infection, vaccination, physical examination, BMI, HAD Scale, or LC symptoms) between the groups (“responders” and “nonresponders”). Regarding the blood analysis requested in the consultation, the “responders” had significantly higher levels of creatinine than the “nonresponders” (Table 5) and also a lower ratio of protein/creatinine (Figure 3). Even though the protein levels were not significantly different between “responders” and “nonresponders”, most likely because of the small sample size, there seems to be a trend for the “responders” to present higher levels. Then, we built a logistic regression model using the ratio of proteins/creatinine and we obtained that if this ratio is bigger than 10, we have an accuracy of 78%, with a sensitivity of 0.75 and a specificity of 0.8. The rest of the analytical variables collected did not show significant differences.

Then, we also analyzed the response to treatment in relation to the presence of obesity and the vaccination state. 

#### 3.4.1. Body Mass Index

We classified patients according to their BMI. Subjects with obesity (>30 Kg/m^2^) had a significantly higher frequency of severe clinical symptoms requiring admission (66.7% vs. 0.0% in patients without obesity, *p* = 0.015). Regarding the values from the RHBC, patients with obesity presented a significantly lower number of meters walked in the 6MWT prior to treatment than patients without obesity (meters in patients without obesity: 461.75 ± 46.80 vs. meters in patients with obesity: 384.20 ± 44.30, *p* = 0.015), lower Borg final dyspnea score (nonobesity patients: 6.50 ± 1.40 vs. patients with obesity: 3.33 ± 1.10, *p* = 0.016) and lower FIS final for fatigue (nonobesity: 7.50 (7.00–8.50) vs. patients with obesity: 4.00 (4.00), *p* = 0.048). In regard to the laboratory values, patients presenting with obesity had higher levels of ALT (patients without obesity: 14.50 (13.00–17.75) vs. those with obesity: 21.00 (17.00–50.50), *p* = 0.045) and GGT (nonobesity: 11.50 (9.25–13.00) vs. obesity: 46.00 (27.50–71.00), *p* = 0.002) than patients without obesity. No significant differences were found regarding the level of physical activity or the HAD scale score.

#### 3.4.2. Vaccination

A total of 78% of the participants had received a first dose of the SARS-CoV-2 vaccine, with the majority receiving the Pfizer vaccine (42.9%). A total of 28.6% of the patients had received a second dose, with the majority receiving the Pfizer vaccine (78%). All vaccinated subjects were vaccinated from 15 January 2021, to 25 August 2021, to prioritize this therapy. Linking vaccination with COVID-19 severity, 27.3% of vaccinated and 33.3% of unvaccinated patients showed severe symptoms and required hospitalization. While the patients vaccinated with a first dose did not present significant differences in the values obtained in the RHBC, the patients vaccinated with the second dose did present a significant difference in the final Borg/FIS value (unvaccinated: 7.50 (7.00–8.50) vs. vaccinated: 4.00 (4.00), *p* = 0.048), being higher in unvaccinated patients than in vaccinated patients.

## 4. Discussion

Long COVID syndrome is a complex multiorgan symptom that develops after an acute COVID-19 infection and persists beyond 12 weeks post-infection. One of the most frequently associated symptoms with the greatest impact on the quality of life of patients is fatigue. Despite having a clearly negative impact on the life and functionality of patients, there is no specific treatment for this pathology, so a holistic approach to patient management is currently recommended. Therefore, the investigation of this pathology and its possible areas of treatment is essential. In the present study, the role that a multidisciplinary rehabilitation program could play in the treatment of LC patients was examined. Moreover, we want to highlight that although some authors have studied predictors of LC appearance [14,21], there are no studies on biomarkers of response to rehabilitation treatment, which would be very useful to determine the best candidates for this treatment. For this reason, our second objective was to investigate possible prognostic biomarkers of the response to rehabilitation treatment in these patients.

In this sense, the novelty of the present study lies in the fact that we have applied a supervised exercise program similar to that described by Cattadori et al. [28] and studied for the first time the possibility of predicting the response to this rehabilitation treatment in patients with LC syndrome and chronic fatigue using basic laboratory, proinflammatory, and autoimmune biomarkers.

First, it should be noted that the baseline characteristics of our cohort were consistent with the higher risk of developing LC described in the literature, such as middle age, obesity, and female sex [3]. It is known that the incidence of LC is influenced by factors, such as female sex, aging, the severity of COVID-19 disease, or the presence of different SARS-CoV-2 variants [29,30]. A total of 64.3% of the patients had an acute COVID-19 condition of mild/moderate severity, and the remaining 35.7% suffered a severe condition requiring hospital admission, with expected alterations [31,32] in the admission analysis (elevated levels above AST, ALT, CRP, D-dimer, and IL-6 reference values). The mean LC symptoms and their distribution were similar to those observed in other studies [8,19,22,33], mostly reporting fatigue, headache, dyspnea, amnesia, and olfactory disturbances. In the laboratory analysis, patients presented an abnormally high mean fibrinogen (445.80 (57.90)) and GSR (14.92 (8.40)), consistent with the analytical profiles associated with LC observed in other studies [34].

After rehabilitation treatment, the participants presented a significant improvement in the meters walked and in oxygen saturation and SBP, with stability in the percentage of meters walked and in the Borg/FIS scales. Taken together, we can consider that the patients in our cohort presented a partial response to the rehabilitation treatment. The results regarding the improvement in meters walked were similar to other studies [35,36]. However, other authors showed an improvement in the Borg/FIS scale score after rehabilitation treatment [35]. Our results may be influenced by the small number of patients since it is a pilot study. Regarding rehabilitation treatment, it is important to highlight that moderate-intensity exercise seems to improve the immune system, so, it should be recommended as a non-pharmacological way to cope with the COVID-19 virus [37].

Regarding obesity, patients with obesity presented a significantly lower number of meters walked in the 6MWT prior to treatment than patients without obesity, as expected. However, those with obesity presented a significantly lower subjective sensation of fatigue or dyspnea after treatment. This could be due to an underestimation of this sensation in obese patients, probably accustomed to a certain basal breathlessness [38].

In relation to the vaccination state, it is important to note that vaccination has the potential to reduce the risk of LC [39]. In our study, patients vaccinated with the second dose against SARS-CoV-2 compared to nonvaccinated patients presented a significantly lower subjective sensation of fatigue. In this sense, some authors have suggested that vaccines could improve LC symptoms by activating T-lymphocytes and eliminating the viral reservoir, causing an increase in the immune response, or diverting the inappropriate autoimmune response present in LC [40].

Significantly lower levels of creatinine were observed in “nonresponders” than in “responder” patients. Moreover, albumin and protein levels were also lower in “nonresponders”, while globulin levels were higher, although without reaching statistical significance. This lack of significance might be due to the small sample size. It must be noted that all values (levels of creatinine, protein, albumin, and globulin) were within normal ranges. Overall, these data seem to indicate that those patients with better protein nutrition will present a better response to rehabilitation treatment, as described [41]. Moreover, some studies [42,43] have linked higher albumin levels and lower globulin levels with a favorable prognosis for COVID-19. Further studies should validate this observation. Additionally, it made us think that perhaps these parameters could predict the response to rehabilitation treatment. In this sense, we studied the correlations of different analytical parameters obtained on the first day of LCMC with the evolution of the disease. The mean result was that the difference in the percentage of meters walked correlated positively with creatinine. Similarly, Robertson et al. recently described some alterations in the molecular composition of long COVID-19 patient urine, detected using Raman spectroscopic/computational analysis [44].

Unlike some authors that have found that inflammatory and autoimmune parameters could be involved in the pathogenesis of LC, we could not find any correlation between the response to treatment with inflammatory classical parameters, only a negative correlation of the difference in FIS for fatigue with the titers of ANA [45,46,47,48,49]. On the other hand, Kruger et al. recently described, in an LC cohort, a reduced level of plasma kallikrein compared to controls, an increased level of platelet factor 4 (PF4) von Willebrand factor (VWF), and a marginally increased level of α-2 antiplasmin (α-2-AP). They suggested that the presence of these proinflammatory molecules could possibly explain why individuals with long COVID suffer from chronic fatigue, dyspnea, or cognitive impairment [50]. In this sense, we found that platelets and fibrinogen were increased in nonresponding patients.

Several limiting aspects must be considered when interpreting the findings of this study. First, as it is a pilot study, the reduced sample size could decrease the statistical power and potentially bias some results. With only nine patients who have completed the treatment, it is important to acknowledge the limitations of generalizability to a larger population. Therefore, the findings from this pilot study are exploratory in nature and should be interpreted as such. Second, the study duration was limited because the research work consumed a lot of healthcare resources and daily care could no longer be interfered with. Third, the results cannot be generalized to the total population due to the lack of representation of male or older patients. Fourth, the nature of the treatment prevented the intervention from being blinded to both the participants and the researchers, and there was no control group for establishing assumptions of causality. Fifth, although the DePaul Symptom Questionnaire is a validated self-report questionnaire of 54 symptoms with excellent discriminant validity to assess and monitor the progression of myalgic encephalomyelitis/chronic fatigue syndrome [51], we decided to use FIS and the Borg scale to assess fatigue and dyspnea, respectively. Sixth, although the walking length increased after the rehabilitation program, this may be partly due to the natural process of the disease. However, a strong point of this study was the number of variables collected and the absence of risks associated with the intervention. However, further studies are needed in order to validate our findings.

## 5. Conclusions

In conclusion, patients with LC and fatigue demonstrated a partial response but were favorable to the rehabilitation intervention. Although the sample size is small, most of the patients were responders to treatment. A good state of protein nutrition was related to a better response to a supervised exercise program. In addition, the results are promising regarding the establishment of creatinine as a possible predictive biomarker of the response to rehabilitation treatment to prioritize and improve patient care. 

## Figures and Tables

**Figure 1 viruses-15-01452-f001:**
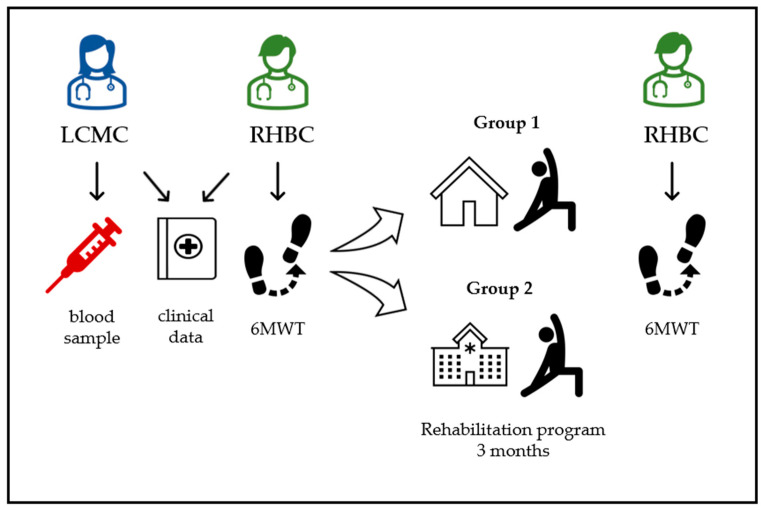
Scheme of the study. LCMC, long COVID monographic consultation; RHBC, rehabilitation consultation; 6 MWT, 6 min walk test.

**Figure 2 viruses-15-01452-f002:**
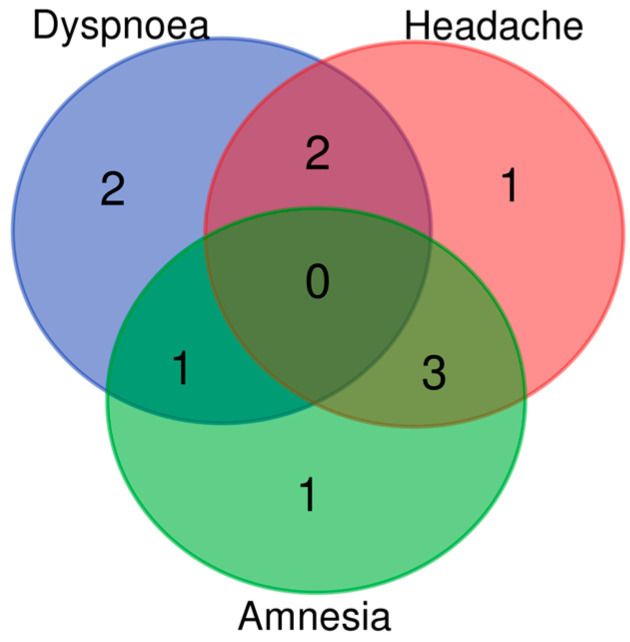
Symptomatology distribution among the study participants (only the most common symptoms are represented).

**Figure 3 viruses-15-01452-f003:**
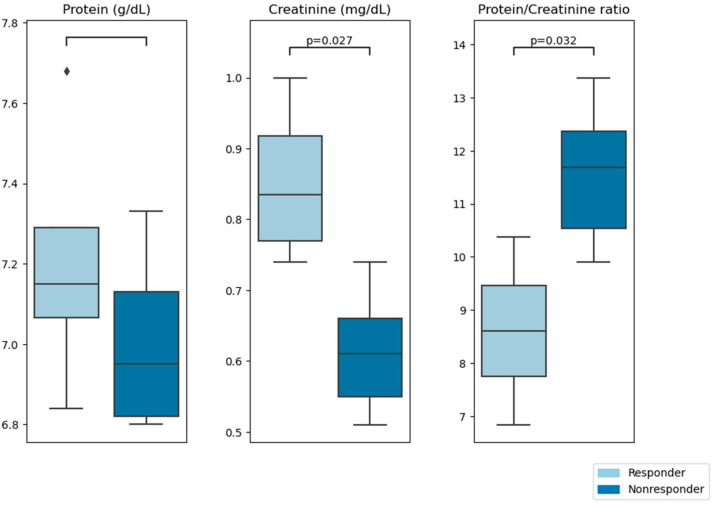
Protein, creatinine, and protein/creatinine ratio between “responders” and “nonresponders”.

**Table 1 viruses-15-01452-t001:** Baseline characteristics and features of acute COVID-19 in our cohort.

Baseline Characteristics of the Patients (n = 14)	Ratio; Mean ± SD; %
Gender (males:females)	0:100
Age (years)	44.21 ± 5.50
Comorbidities:	35.7%
Hypertension	14.3%
Type 2 diabetes mellitus	14.3%
Dyslipidemia	21.4%
Cardiopathy	7.1%
Pneumopathy	7.1%
Allergy	21.4%
BMI:	
Normal weight	7.1%
Overweight	50%
Obesity I	21.4%
Obesity II	21.4%
Physical activity:	
Low	78.6%
Moderate	21.4%
Acute COVID-19 severity:	
Mild/moderate	64.3%
Severe	35.7%
Hospital admission	35.7%
Vaccinated	78.6%
Single dose	28.6%
Booster dose	50.0%
ESR	28.6%
Abnormal chest X-ray	35.7%
Treatment:	
Symptomatic	100%
Antibiotic	42.9%
Corticoids	35.7%
LMWH	35.7%
Remdesivir	7.1%

Data represented as ratio, means (± standard deviation, SD), or percentage. BMI, body mass index; ESR, erythrocyte sedimentation rate; LMWH, low molecular weight heparin.

**Table 2 viruses-15-01452-t002:** Laboratory results during hospital admission and in the LCMC.

Variables	Analysis at Admission (n = 5) Mean ± SD	Analysis in the LCMC (n = 14) Mean ± SD	*p*-Value
Creatinine (mg/dL)	0.61 ± 0.10	0.69 ± 0.10	0.756
Albumin (g/dL)	3.73 ± 0.19	4.20 ± 0.41	0.861
Globulin (g/dL)	2.59 ± 0.29	2.89 ± 0.51	0.444
Protein (g/dL)	6.33 ± 0.30	7.09 ± 0.31	0.013
AST (U/L)	71.40 ± 68.30	21.61 ± 7.40	0.216
ALT (U/L)	89.80 ± 78.20	21.46 ± 17.10	0.199
Hemoglobin (g/dL)	13.4 ± 1.10	13.3 ± 1.20	0.294
Leucocytes (×10^9^/L)	5.41 ± 2.20	6.10 ± 1.50	0.974
Lymphocytes (×10^9^/L)	1.36 ± 0.30	1.95 ± 0.80	0.179
CRP (mg/dL)	2.48 ± 2.00	0.48 ± 0.22	0.193
D-dimer (ng/dL)	759.50 ± 458.90	338.0 ± 158.10	0.473
IL-6 (pg/dL)	24.26 ± 14.60	3.13 ± 1.10	0.295

Data represented as means (± standard deviation, SD). LCMC, long COVID monographic consultation; AST, aspartate aminotransferase; ALT alanine aminotransferase; CRP, C-reactive protein; IL-6, interleukin 6. Significant differences between values (*p* < 0.05).

**Table 3 viruses-15-01452-t003:** Clinical and analytical characteristics of the patients in the LCMC.

Constants	Mean ± SD; Ratio; %; Median (25th–75th) (n = 14)
Time between first diagnosis and LCMC (weeks)	28.36 ± 15.57
SBP (mmHg)	128.00 ± 13.10
DBP (mmHg)	83.83 ± 11.20
Peripheral oxygen saturation (%)	97.82 ± 1.20
Cardiac frequency (bpm)	88.0 ± 32.10
Breathing frequency (N:P, %)	100:0
Physical exploration (N:P, %)	21:78
Cardiorespiratory auscultation	86:14
Abdominal examination	93:7
Neurologic examination	100:0
Mood–emotional state:	
HAD anxiety scale:	
Case	57.1%
Borderline case	21.4%
No case	21.4%
HAD depression:	
Case	14.3%
Borderline case	42.9%
No case	42.9%
Blood analysis:	
Lymphocytes (×10^9^/L)	1.61 (1.40–2.55)
Inflammatory markers	84.6%
Autoimmunity	16.7%
SARS-Cov-2 serology	100%

Data represented as means (± standard deviation, SD), ratio, median (25th–75th percentiles), or %, according to convenience. SBP, systolic blood pressure; DBP, diastolic blood pressure; HAD Hospital Anxiety and Depression Scale.

**Table 4 viruses-15-01452-t004:** 6MWT, Borg, and FIS data before and after rehabilitation treatment.

Variables	Before RHB Treatment (n = 9)	After RHB Treatment (n = 9)	*p*-Value
SBP (mmHg)	129.92 ± 20.50	124.78 ± 23.50	0.009
Cardiac frequency (bpm)	100.54 ± 17.70	107.44 ± 18.60	0.157
Oxygen saturation (%)	98.23 ± 1.20	98.33 ± 1.30	0.110
Borg dyspnea scale	4.54 ± 2.60	5.44 ± 2.00	0.716
Borg fatigue scale	6.14 ± 1.30	6.89 ± 1.90	0.364
Meters walked	431.92 ± 58.90	439.33 ± 62.30	<0.001
Percentage of meters walked	78.5 ± 9.90	76.50 ± 6.30	0.077

Data represented as means (±standard deviation, SD). Significant differences between the initial and final values (*p* < 0.05). SBP, systolic blood pressure; FIS, Fatigue Impact Scale; RHB, rehabilitation. Oxygen saturation measured on room air.

**Table 5 viruses-15-01452-t005:** Classification of patients who followed the hospital rehabilitation program according to the improvement in walked meters and oxygen saturation percentage as “responders” and “non-responders”.

Variables	Responders (n = 4)	Nonresponders (n = 5)	*p*-Value
Hemoglobin (g/dL)	13.27 ± 1.90	13.12 ± 0.40	0.882
Leucocytes (×10^9^/L)	6.72 ± 1.92	6.27 ± 1.40	0.713
Lymphocytes (×10^9^/L)	1.87 (1.31–2.46)	1.43 (1.37–2.08)	0.730
Platelets (×10^9^/L)	253.75 ± 28.50	310.60 ± 34.70	0.031
Creatinine (mg/dL)	0.85 ± 0.10	0.61 ± 0.10	0.017
Albumin (g/dL)	4.38 ± 0.27	3.93 ± 0.50	0.192
Globulin (g/dL)	2.82 ± 0.19	3.07 ± 0.66	0.531
Protein (g/dL)	7.20 ± 0.30	7.01 ± 0.20	0.331
AST (U/L)	20.5 ± 2.80	18.00 ± 4.00	0.313
Fibrinogen (mg/dL)	391.50 ± 47.70	486.60 ± 43.20	0.020
D-Dimer (ng/dL)	447.75 ± 241.80	321.40 ± 84.60	0.333
Ferritin (ng/dL)	29.50 ± 21.70	38.40 ± 18.70	0.540
IL-6 (pg/dL)	2.70 (2.70–5.47)	2.70 (2.70–3.65)	0.999
CRP (mg/dL)	0.40 (0.40)	0.40 (0.40–0.85)	0.413
ESR (mm)	11.00 ± 8.70	21.40 ± 7.60	0.110
Ab SARS-CoV-2 (index)	36.21 ± 20.90	69.21 ± 34.60	0.123
Autoimmunity (+%)	25%	20%	0.999

Data represented as means (±standard deviation, SD), median (25th–75th percentiles), or %, according to convenience. Marked in bold the significant differences between groups (*p* < 0.05). Ab SARS-CoV-2, SARS-CoV-2 antibodies; AST, aspartate aminotransferase; ESR, Erythrocyte Sedimentation Rate; IL-6, interleukin 6; CRP, C-reactive protein.

## Data Availability

Not applicable.

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
