# Peer review of "Clinical and Biomarker Profile Responses to Rehabilitation Treatment in Patients with Long COVID Characterized by Chronic Fatigue"

_viruses, 2023, doi:10.3390/v15071452_

Round 1

Reviewer 1 Report

The authors’ paper suggests the importance of rehabilitation program to improve fatigue due to long COVID-19.  This issue is clinically and socially important; however, several revision points are necessary for further consideration.

Major points:

1)     The appropriateness of the contents of the rehabilitation program is unclear.  The process of program design should be clarified.

2)     Although it is stated that the walking length increased after the rehabilitation program, it is not necessarily due to the rehabilitation, but may be partly due to the natural process of improvement.  Adding some data from follow-up of patients who did not require intervention, if available, would resolve this concern.

3)     The effects of other medication and related supplements should also be excluded in the process of improvement.

4)     Regarding the comparison of Group 1 and Group 2, it is somewhat arbitrary to focus only on BMI and vaccination history as patient factors.  More data should be provided on what the other factors were.

5)     As a whole study, the incorporated sizes of the long COVID patients were quite small.  Authors should address the limitation issues raised in this study.

6)     The mechanism and the rationales of lowered creatinine and protein in the "non-responders" should be discussed.  How about the other significant makers such as platelets and fibrinogen levels?

Minor points:

1)     Respiratory function tests were not performed before or after rehabilitation?  The presence or absence of respiratory failure is very important data.

2)     Was the patient instructed to try home exercises?  If so, this is one of the interventions and should be mentioned in the text.

3)     The number of N should be listed in each Table.  Especially for Table 2, it is essential as the number of patients may vary by column.

Reviewer 2 Report

Kindly find the attached comments.

The study requires minor grammar editing.

Reviewer 3 Report

1.Title: Should be revised for clarity. I suggest: “Clinical and biomarker profile responses to rehabilitation treatment in patients with long COVID characterized by chronic fatigue.”

2.P. 2, top: It is stated that long COVID is defined as persistence of symptoms >4 weeks after acute COVID-19. This is the U.S. CDC definition; the WHO requires persistence >12 weeks. This should be noted (can use ref.10.) In addition, in the first sentence of the Discussion the authors use the >12 week duration to define long COVID. Please clarify this discrepancy.

3. P. 2, 2nd paragraph: A single citation (ref. 12) is provided to present the multiple phenotypes encompassed by long COVID. Additional recent references should be provided reviewing the complexity of the diagnosis and its pathophysiology, which likely drives different clinical phenotypes (e.g., Ahamed J, et al. J Clin Invest 2022;132:e161167; Nalbandian A, et al. Nature Med 2021;27:601; and Turner S, et al., Trends Endocrin Metab 2023;34:321).

4.The tables are confusing as it is unclear how many subjects were included in each table. Please provide the n for each table.

5.Table 1: Spell out “VSG” and “LMWH.” Add the % vaccinated against SARS-CoV-2. Since much is made about protein levels being a driver of response, please include total protein, albumin and globulin levels here.  

6. Table 3: Add the mean and range for number of weeks post-acute-COVID-19. This relates to comment no. 2 about definitions of long COVID.

7. P. 5, middle: Of the 28.6% of participants hospitalized, did any require ventilator support or high-flow O2? If not, what were the reasons for their hospitalizations?

8. Table 2 presents data on the 5 subjects requiring hospitalization for acute COVID-19. What is the denominator? Earlier it is stated that 28.6% required hospitalization so it is not 10 or 14.

9. Figure 2 presents limited information that could simply be given in the text. Alternatively,  it can be redrawn as a Venn diagram showing interactions among the three non-fatigue parameters: headache, dyspnea, and amnesia.

10. Please state how amnesia was diagnosed.

11. Table 3: The peripheral O2 sat was WNL here at 98%, I presume on room air. But it is stated on p. 8 that these values “presented significantly higher than compared to the initial values.” This is a critical piece of information. Both pre- and post-O2 sat values should be presented in at least one of the tables.

12. I am surprised that with such a small number of subjects the difference in O2 sat between before and after therapy groups--only a 0.1% difference--was significant at a p of 0.01. Is that correct?

13. Table 5: Why are only 9 rather than 10 or 14 subjects presented here? As noted in comment no. 4, please provide the n for each table with an explanation about why it differs among tables.

14. Table 5: “Autoimmunity” is listed at 20%. I assume this is based on the two “positive ANAs” and one “positive AcL” noted in the text. If so, please mention this in the legend and include in the text the titers for the ANAs and the antibodies and their titers measured to determine the AcL. Also and note whether they were present pre-COVID-19 or at the time of COVID-19 diagnosis.  

15. Table 5: I am unfamiliar with the term “Ac SARS-CoV-2 (index).” Please explain in the text.

16. Discussion: Should note that others have published exercise training regimens to ameliorate long COVID (e.g., Cattadori G, et al. J Clin Med 2022;11:2228).

17. Discussion and Conclusion: If the authors wish to claim that changes in creatinine and protein reflect differences in nutrition and might underlie response to rehabilitation therapy, then more information needs to be presented. Creatinine changes could relate to differences in hydration or resolution response times to an acute-COVID-19-related kidney injury. As noted in comment no.5, it is important to document if changes in albumin vs. globulin levels are responsible for any differences seen in total protein levels, with values provided in one of the tables.

Round 2

Reviewer 1 Report

The revision was appropriately performed.

Author Response

Thank you for your review.

Reviewer 2 Report

The paper has been extensively revised and is now acceptable for publication.

Author Response

Thank you for your review.

Reviewer 3 Report

The authors have responded adequately to all of the issues raised in my original critique except for one, which must be addressed:

1.Table 2 and text: as requested the authors have included total protein and albumin levels showing no statistical difference between albumin levels but a difference in TP levels. That suggests the component of TP leading to the statistically significant difference was related to globulin, a value I had also requested but was not included. Indeed, simple arithmetic shows that pre- globulin was 2.60 but post it was 2.89. Please include these  values in Table 2 and do the statistics.

2. Assuming I am correct in comment no. 1, the authors then need to discuss this result. They should quote studies showing that an increase in albumin is a good prognostic marker in COVID-19 (e.g., Xu Y, et al. Int J Gen Med 2021;14:2785). But an increase in globulin could be a bad sign if it reflects a hyper-inflammatory state.

3. Table 5 and text: Same issue as in 1 and 2. Please provide the globulin values and the statistics and add a note in Discussion about those results, if significant. (Given the small numbers, as noted by the authors in this section, it may not be.)
